# Oxford Nanopore Technology-Based Identification of an *Acanthamoeba castellanii* Endosymbiosis in Microbial Keratitis

**DOI:** 10.3390/microorganisms12112292

**Published:** 2024-11-12

**Authors:** Sebastian Alexander Scharf, Lennart Friedrichs, Robert Bock, Maria Borrelli, Colin MacKenzie, Klaus Pfeffer, Birgit Henrich

**Affiliations:** 1Institute of Medical Microbiology and Hospital Hygiene, Medical Faculty, Heinrich Heine University Düsseldorf, Moorenstrasse 5, 40225 Duesseldorf, Germany; lennart.friedrichs@med.uni-duesseldorf.de (L.F.); colin.mackenzie@med.uni-duesseldorf.de (C.M.); klaus.pfeffer@hhu.de (K.P.); 2Department of Ophthalmology, Medical Faculty and University Hospital Duesseldorf, Heinrich Heine University Düsseldorf, Moorenstrasse 5, 40225 Duesseldorf, Germany; robert.bock@med.uni-duesseldorf.de (R.B.); maria.borrelli@med.uni-duesseldorf.de (M.B.)

**Keywords:** whole-genome sequencing (WGS), eye microbiome, *Acanthamoeba*, endosymbiont, keratitis, eye infection, *Acanthamoeba* keratitis

## Abstract

(1) Background: Microbial keratitis is a serious eye infection that carries a significant risk of vision loss. *Acanthamoeba* spp. are known to cause keratitis and their bacterial endosymbionts can increase virulence and/or treatment resistance and thus significantly worsen the course of the disease. (2) Methods and Results: In a suspected case of Acanthamoeba keratitis, in addition to *Acanthamoeba* spp., an endosymbiont of acanthamoebae belonging to the taxonomic order of Holosporales was detected by chance in a bacterial 16S rDNA-based pan-PCR and subsequently classified as Candidatus *Paracaedibacter symbiosus* through an analysis of an enlarged 16S rDNA region. We used Oxford Nanopore Technology to evaluate the usefulness of whole-genome sequencing (WGS) as a one-step diagnostics method. Here, *Acanthamoeba castellanii* and the endosymbiont Candidatus *Paracaedibacter symbiosus* could be directly detected at the species level. No other microbes were identified in the specimen. (3) Conclusions: We recommend the introduction of WGS as a diagnostic approach for keratitis to replace the need for multiple species-specific qPCRs in future routine diagnostics and to enable an all-encompassing characterisation of the polymicrobial community in one step.

## 1. Introduction

Microbial keratitis has an annual incidence of approx. 1.5–2 million cases in developing countries [1,2]. It leads to corneal opacity, which is one of the main causes of corneal blindness worldwide [1,2]. The most common risk factor in approximately 70% of all cases is the wearing of contact lenses [3]. Bacterial infections of the corneal tissue account for 90% of all microbial keratitis cases [3,4], *Aspergillus* spp., *Fusarium* spp. and *Candida albicans* [5], as well as viruses such as herpes simplex virus [1], are further pathogens. *Acanthamoeba* keratitis is rare but particularly worrying, as it is associated with severe eye damage [6]. *Acanthamoeba* spp. are protozoans that are naturally found in dust, soil, and water, and the contamination of contact lenses usually occurs through unhygienic cleaning methods [1]. Infections are difficult to diagnose and to treat. Due to the relative rarity of *Acanthamoeba* keratitis compared to other causes of keratitis (bacterial, fungal, viral), it is often misdiagnosed, especially in the early stages of the disease [7]. If not diagnosed and left untreated, *Acanthamoeba* keratitis can lead to irreversible vision loss; if recognised early, it can be cured by treatment with a combination of biguanide and aromatic diadine antibiotics given for several months [6].

The survival, virulence, and antibiotic resistance of corneal *Acanthamoeba* can be enhanced in the presence of endosymbiotic bacteria [8,9,10,11], which persist intracellularly and are transmitted within the protozoan [12]. Among these endosymbionts are well-known pathogens, such as *Legionella pneumophila*, *Coxiella burnetii*, *Pseudomonas aeruginosa*, *Helicobacter pylori*, *Cryptococcus neoformans*, and *Chlamydia trachomatis* [13,14], as well as less-known genera of the order Holosporales, such as Candidatus *Caedibacter* and *Paracaedibacter*, which have emerged as potential pathogens in corneal infections [8,9,10,11]. *Acanthamoeba* can also affect endosymbiont virulence and invasiveness by protecting it from antibiotic treatment and providing good growing conditions. In addition, the interaction between host and endosymbiont may exacerbate the course of keratitis due to the presence of pro-inflammatory bacterial compounds [15].

For the diagnosis of microbial keratitis, a corneal culture remains the routine standard procedure; however, this approach is impaired by long diagnostic turnaround times and low sensitivity [1,2]. Confocal corneal microscopy is significantly faster compared to culture but with little differentiation between bacterial and fungal keratitis [16]. Molecular genetic methods, like species-specific qPCR or broad-range qPCR-based Sanger sequencing, have already shown convincing results in the diagnosis of keratitis [2]. Recent studies demonstrated PCR as a suitable tool for the diagnosis of *Acanthamoeba* keratitis, with a sensitivity of 70–100% and a specificity of 90–100% [3,17,18,19,20,21]. The limitation of PCR is the diagnostic restriction of the primers used; the more specific the primers are, the greater the risk that pathogen variants will be missed [2].

In recent years, several attempts have been made to define the ocular microbiome, but this has not yet been well-characterised [22]. While, in the past, culture-based methods were mainly used to analyse the ocular microbiome, through which only a few microorganisms were identifiable, more recently, the focus has been on 16S rDNA-based next generation sequencing (NGS) methods, which, in contrast to culture-based methods, has revealed a rich bacterial ocular microbiome, mainly consisting of staphylococci, streptococci, propionibacteria, and micrococci [22,23,24].

Metagenomics/shotgun whole-genome sequencing (WGS), would enable the detection of unexpected pathogens by identifying the entire microbial DNA in a sample. This means that all bacteria, fungi, DNA viruses, and parasites can be detected in a single analysis. Unfortunately, WGS has not yet become established in the routine diagnostics of microbial keratitis because of the high costs and long processing times, although its potential to detect the microbiome including pathogens and their endosymbionts could revolutionise our understanding of microbial interactions in ocular infections [2].

In this pilot study, we take the first steps to validate a diagnostic application for WGS for pathogen detection in *Acanthamoeba*-associated keratitis and compare WGS with the routine standard methods of confocal corneal microscopy and PCR.

## 2. Materials and Methods

### 2.1. DNA Extraction

The corneal biopsy was subjected to proteinase K digestion at 56 °C for 1 h in 200 µL of G2 buffer (Qiagen, Hilden, Germany). Thereafter, the DNA was purified in an EZ1 advanced machine using DNeasy Blood & Tissue Kit (Qiagen, Hilden, Germany) according to the manufacturer’s instructions for Gram-positive bacteria. DNA eluate (100 μL) was stored at −20 °C until use.

### 2.2. qPCR

The *inhouse* Acanthamoeba qPCR was conducted using primers and probe targeting a 180 bp region of the 18S rRNA gene (listed in Appendix A, Sheet Primers and Probes), as published by Qvarnstrom et al. [25] and in a total volume of 25 μL, comprising 1× No ROX qPCR MasterMix (Eurogentec, Seraing, Belgium), 0.3 µM of each primer, 0.2 µM of the probe, and 2.5 µL of the DNA sample. The cycling profile encompassed initial denaturation at 95 °C for 2 min, followed by 40 cycles (95 °C for 15 s, 60 °C for 1 min).

Bacterial pan-PCRs comprising the variable 16S rDNA regions V1–V2 (nt 27–591) and V1-V4 (nt 27–907) were conducted in a total volume of 25 μL, as published previously [26]. To decode 1400 bp of the 16S rDNA, overlapping PCR products (nt 27–1492 and 799–1492) were amplified using additional primers 16S-F3 and 16S-R3 (listed in Appendix A, Sheet Primers and Probes). The cycling profile encompassed the initial denaturation at 95 °C for 2 min, followed by 40 cycles (95 °C for 30 s, 55 °C for 30 s, 72 °C for 1.5 min) and a melting curve analysis. PCRs were all run on a Thermocycler CFX 96 Touch (Bio-Rad, Hercules, CA, USA). Sanger sequencing of the PCR products was performed by the Biological Medical Research Centre of the Heinrich-Heine University, Duesseldorf. The most homologues species were identified in BLASTN analysis of NCBI (https://blast.ncbi.nlm.nih.gov/Blast.cgi?PROGRAM=blastn&PAGE_TYPE=BlastSearch&LINK_LOC=blasthome accessed on 4 July 2024).

### 2.3. Whole Genome Sequencing

Oxford Nanopore Technology-based whole-genome sequencing (WGS) works by passing single DNA molecules through nanopores with a constant ionic current. The speed of translocation is controlled by a motor protein that stepwise pushes the nucleic acid molecule through the nanopore. Specific changes in the ion current during translocation correspond to the nucleotide sequence (A, C, G or T) in the sensor region and are decoded using computer algorithms [27]. A DNA library was prepared for sequencing using one barcode and the Native Barcoding Kit 24 V14 (SQK-NBD114.24; Oxford Nanopore Technologies, Oxford, UK). The library was sequenced using the PromethION with R10 version nanopores (Oxford Nanopore). We have uploaded the Nanopore sequencing data to NCBI (BioProject ID: PRJNA1175606). After sequencing, raw data were base-called using Dorado 7.3.11, and reads with lengths of ≥ 1000 bp were mapped to the comprehensive database that we compiled with sequences from the NCBI, which contained 11,019 bacterial, 154 eukaryotic, 474 archaeal, and 13,867 viral species (see Appendix A, Sheet *KrakenDatabase Version* 19 June 2024) using Kraken 2.1.1 [28]. Minimap 2.17 [29] was used for back-mapping to the genomes of *Acanthamoeba castellanii* strain Neff (accession number 1257118), Candidatus *Paracaedibacter symbiosus* (acc-no. 244582), *Arthrobacter* sp. KBS0702 (acc-no. 2578107), *Plasmodium vivax* (acc-no. 5855), and *Homo sapiens* (acc-no. 9606).

## 3. Results

### 3.1. Case Report

A 46-year-old male patient presented to the Department of Ophthalmology at the University Clinic of Duesseldorf complaining of eye pain for three months with significant worsening over the last 3 days, with reddening of the left eye, epiphora, photophobia, blepharospasm, and significant visual deterioration. He had been experiencing eye pain for three months and had been using soft contact lenses up to that point.

The examination of the right eye showed normal age-appropriate findings, whereas the left eye showed a clear conjunctival injection with chemosis and corneal decompensation. In the centre of the cornea was a dense, large ring infiltrate with a surface defect, which made a more precise assessment of the anterior eye chamber impossible. A hypopyon was not detected. Sonography showed no evidence of vitreous involvement.

An eye swab with subsequent routine diagnostic in-house qPCR for the detection of Acanthamoeba and a bacterial pan-PCR were positive and revealed Acanthamoeba spp. The patient was hospitalised, and inpatient therapy was performed with propamidine isoethionate, polyhexanide, and atropine eye drops for Acanthamoeba keratitis. As there was no improvement after one day, topical voriconazole was added. Confocal corneal microscopy showed the characteristic cysts compatible with acanthamoebae.

During the clinical stay, the irritation of the left eye improved. The patient reported a subjective decrease in pain and blepharospasm. The infiltrate became more clearly demarcated, and the epithelial defect showed a regressive tendency. As the corneal oedema decreased, endothelial precipitates and irritation of the anterior chamber became progressive. As adjuvant therapy, corneal crosslinking (riboflavin eye drops in combination with UVA irradiation) was performed. As the clinical presentation continued to improve, a penetrating keratoplasty à chaud was performed as an inpatient procedure for germ reduction. A corneal biopsy was sent to the Institute of Medical Microbiology and Hospital Hygiene. Postoperatively, there were no complications after keratoplasty with an improvement in visual acuity. The patient was discharged three days after keratoplasty with significantly reduced therapy with polyhexanide and dexamethasone and no evidence of a recurrence of the infection. Close monitoring was carried out up to four months postoperatively, at which time no further pathology was observed.

### 3.2. Microbiological Diagnostics

At the Institute of Medical Microbiology and Hospital Hygiene, the genomic DNA of the corneal biopsy was positive for genus-specific Acanthamoeba-PCR. A pan-bacterial 16S rDNA PCR was performed, and Sanger sequencing of the 16S rDNA amplicon (V1–V3 region) led to the identification of a bacterium belonging to the order Holosporales, members of which are known endosymbionts of Acanthamoeba spp. To narrow down the bacterial species, further bacterial Pan-PCRs were carried out amplifying and subsequently decoding 1.4 kb 16S rDNA region in total. A multiple-sequence alignment of the query to 16S rDNA homologue regions in known Holosporales genomes (see Appendix A, Sheet Holosporales Species) enabled the construction of a phylogenetic tree (see Figure 1), which suggests that the patient’s endosymbiont belongs to Candidatus *Paracaedibacter symbiosus*.

To validate the power of WGS as a one-step molecular genetic diagnostic approach, we sequenced the DNA sample from the keratoplasty biopsy using Oxford Nanopore Sequencing Technology (ONT). A bioinformatic analysis was performed using the k-mer-based approach of Kraken2. A total of 3,203,564 reads (99.92%) were classified as *Homo sapiens*, 1683 reads (0.05%) were classified as *Acanthamoeba castellanii* (strain Neff), 108 reads (<0.005%) were classified as *Arthrobacter* spp. KBS0702, 35 reads (<0.005%) were classified as *Plasmodium vivax*, and 32 reads (<0.005%) were classified as Candidatus *Paracaedibacter symbiosus* (see Table 1).

A Minimap2-based remapping of the reads to the respective genome sequences was used to verify the results (Table 2). The identification of Candidatus Paracaedibacter symbiosus and Acanthamoeba castellanii was confirmed by the verification step. A total of 90.6% of the Candidatus Paracaedibacter symbiosus reads and 55% of the Acanthamoeba castellanii reads were mapped to the respective reference genomes.

However, 49% of the Acanthamoeba reads were mapped to Homo sapiens, indicating that some of the Homo sapiens reads had been incorrectly classified as Acanthamoeba. Misclassification by Kraken2 became also obvious for Arthrobacter spp. and Plasmodium vivax reads. In particular, the supposed Arthrobacter spp. reads were all mapped to Homo sapiens, and the Plasmodium vivax reads were mapped back to both Homo sapiens (100%) and Plasmodium vivax (97.1%). This result suggests contamination of the reference genomes in the data bank (see discussion) and implies that a bioinformatic verification step of WGS data is mandatory.

For the genus Acanthamoeba, 23 different genotypes (type T1 to T23) had been identified, as defined by differences in the 18S rDNA gene, with T4 as the most frequently identified genotype in AK cases [30]. To determine which genotype the detected Acanthamoeba was assigned to, we sequenced the 180 bp 18S rDNA-based Acanthamoeba-PCR product of the routine diagnostics, which covered the 5′ end of the JDP1–JDP2 targeted region (that is commonly used for genotyping) [31]. Additionally, we mapped the Nanopore sequence reads to the 18S rDNA gene of Acanthamoeba castellanii strain Neff and extracted the three overlapping reads. The consensus sequence of the PCR product and the Nanopore reads of 16,171 bp covered the whole genotyping region JPD1–JDP2 (see Figure 2). A Blast analysis of this sequence was conducted. Of the possible 23 genotypes, only the T4 sequence was found.

## 4. Discussion

In addition to the specific detection of *Acanthamoeba* spp. by qPCR at the genus level, an endosymbiont of acanthamoebae was detected rather by chance in the bacterial 16S rDNA-based pan-PCR in a case of microbial keratitis. Both the *Acanthamoeba* and its endosymbiont could simultaneously be classified in a single-step technique by Nanopore-based WGS on the species level as *Acanthamoeba castellanii* and Candidatus *Paracaedibacter symbiosus*. The detection of both microbes from a clinical specimen suggests an endosymbiotic relationship.

### 4.1. Implications of Endosymbiosis for Pathogenicity

Stable associations of bacteria with amoebae leading to long-term symbiotic interactions were described for members of four evolutionary lineages within the *Alphaproteobacteria*, *Betaproteobacteria*, *Bacteroidetes*, and *Chlamydiae* [32]. Endosymbionts, such as *Legionella*, *Chlamydia*, and *Paracaedibacter*, are frequently associated with *Acanthamoeba* [15,32]. Different studies have shown that the detection of these bacterial endosymbionts can provide critical clues to diagnose *Acanthamoeba* infections [15,33,34]. In a case report from Southern Brazil, the detection of *Acanthamoeba*-associated keratitis in soft contact lens wearers revealed the presence of the endosymbiont Candidatus *Paracaedibacter acanthamoeba*, emphasising the role of this microorganism in the pathogenesis of *Acanthamoeba* infections [35]. This was also shown by Cohen et al. (2011), who reported on amoeba-associated keratitis with simultaneous presence of the endosymbiont Candidatus *Odyssella* sp. [36].

The relationship between *Acanthamoeba* and its endosymbionts has significant implications for pathogenicity. The presence of endosymbionts can enhance the virulence of *Acanthamoeba* [8]. The ingestion of *Legionella pneumophila* by Acanthamoeba induced specific gene expression profiles that could enhance amoeba survival and pathogenic potential [10,11]. Rayamajhee et al. (2022) emphasised the role of *Acanthamoeba* as an environmental phagocyte that enhances the survival and transmission of human pathogens, thereby acting as a “melting pot” for microbial interactions [37]. This symbiotic relationship not only favours the persistence of pathogens, as demonstrated for *Helicobacter pylori* [38], but makes the *Acanthamoeba* infection more challenging to treat. For this reason, the detection of both *Acanthamoeba* and the endosymbiont is necessary for adequate therapy. The nanopore-based WGS of clinical specimens combined with bioinformatic analysis will be a suitable approach to verify this hypothesis in the future.

### 4.2. Comparison of Detection Methods

To date, three established detection methods for *Acanthamoeba* in keratitis have been discussed in the literature: culture of corneal scrapings, in vivo confocal microscopy, and PCR. Culture is the most conventional method that has been considered the gold standard. However, it was shown to lack sensitivity and require a long cultivation period of about 2 weeks, which drastically prolongs the time to diagnosis, and other co-pathogens and/or endosymbionts remain undetected or require additional (cultivation) methods [39,40,41]. Confocal corneal microscopy is significantly faster but cannot easily differentiate between bacterial and fungal keratitis [16]. PCR has been frequently used for the detection of acanthamoebae due to its high sensitivity and specificity and broad range of performance, as demonstrated by the differentiation of *Acanthamoeba* genotypes [42]. When comparing the three methods, different research groups came to different conclusions. Yera et al. compared culture and PCR and found that culture and a single PCR test can lead to a misdiagnosis of *Acanthamoeba* keratitis, but a combination of different PCR tests could increase the diagnostic sensitivity [43]. Two other groups compared in vivo confocal microscopy (IVCM), culture, and PCR. They both came to the conclusion that IVCM has the highest sensitivity and specificity in detection [39,44]. However, while Hoffman et al. still considered culture to be the gold standard, Goh et al. instead recommended IVCM or PCR, as it is more readily available compared to IVCM. PCR was also the recommended method compared to culture in a further publication, with which the authors were able to reduce the detection time from approx. 2 weeks to 3 h [40]. The confocal microscopy performed by the Department of Ophthalmology at the University Clinic of Duesseldorf clinic had already delivered an initial indication of *Acanthamoeba* keratitis. On the basis of this clinical suspicion, PCR was carried out at the Institute of Microbiology and Hospital Hygiene for the detection of *Acanthamoeba*.

As PCR is based on specific primer sets, new or genetically divergent strains can be missed, which limits the usefulness of PCR for a comprehensive analysis. The use of broad-range PCRs, as performed in this report, requires a highly time-consuming and resource-intensive extension of the Pan-PCR and subsequent Sanger sequencing to narrow down the bacterial species; the qPCR-based detection of *Acanthamoeba* spp. also yielded no identification of the amoebal species. Depending on the species, acanthamoebae are categorised into different genotypes, which have different potentials for the patho-aetiology of keratitis and encephalitis [45,46].

The use of WGS as a diagnostic avenue, which allows for the sequencing of any DNA contained in a clinical specimen, has proven successful in this report as it revealed a comprehensive picture of the microbial composition, *Acanthamoeba* and endosymbiont, with a one-step diagnostic procedure. 16S rDNA-based NGS has been used to identify endosymbionts of *Acanthamoeba* [15,47]; in 2021, Low and colleagues published a workflow for the detection of keratitis-associated bacteria using this method [48]. To the best of our knowledge, the report presented here is one of the first descriptions of using WGS instead of 16S rDNA-based methods for the comprehensive determination of pathogens and other microorganisms in a keratitis sample.

Before the impression is created that WGS will replace routine qPCR diagnostics tomorrow, we would like to discuss some limitations of WGS. As mentioned above, WGS was shown to present significant challenges, particularly in the accurate classification of *Acanthamoeba* and its endosymbiont, and it harbours the risk of false positive detections due to sequencing artefacts and contaminated genome sequences in data repositories. In our case, some *Homo sapiens* sequences were misidentified by Kraken2 as *Arthrobacter* spp. due to the presence of a highly repetitive sequence element (5′-TGGC(T)-3′) in the genome, and they were identified as false positives by remapping the respective reads with Minimap2 against both the *Homo sapiens* and the *Arthrobacter* reference genome. This issue was already highlighted in a recent study, in which the mapping-based tool Minimap2 partly showed better accuracy in metagenomics classification than the k-mer-based classification tool Kraken2 (although the running times were significantly longer [49]). The cross-contamination of reference sequences can be best explained by the intracellular nature of some pathogens, such as Plasmodia, and subsequent “contamination” of the sequence databases with the respective sequences. The sequencing of intracellular organisms always harbours the risk of host contamination and vice versa. This can lead to the erroneous detection of unrelated intracellular organisms, such as *Plasmodium*, due to sequence “contamination” from human host cells. This problem underscores the importance of using well-curated reference databases to avoid misinterpretation of sequencing data, as Chrisman and co-workers emphasised in their comparative analysis of the human “contaminome”, which describes the contamination of bacterial or viral sequences in whole-genome datasets [50].

### 4.3. Concluding Considerations

The ocular microbiome has not yet been well-characterised [22]. In the past, mainly culture-based methods were used to analyse the ocular microbiome, which was generally characterised by a very low bacterial load. A study by Peter et al. showed that there are approximately 150-fold fewer bacteria on a healthy conjunctiva than on the facial skin [51]. Recently, however, studies based on 16S rDNA NGS have shown that the “healthy” eye is indeed colonised by bacteria, although their biomass is very low [22,23,24]. The healthy ocular surface microbiome mainly consists of Gram-positive genera, such as coagulase-negative staphylococci, streptococci, propionibacteria, and micrococci [23]. Various bacteria, fungi, viruses, and parasites, like *Acanthamoeba*, have already been described to cause keratitis [52,53,54,55]. In 23–55% of cases of *Acanthamoeba* keratitis, these pathogens often occur as co-infections, particularly in connection with traumatic injuries, which represents the entry point for the pathogens [56,57,58,59]. In the WGS analysis reported here, we were able to detect *Acanthamoeba castellanii* and the endosymbiont Candidatus *Paracedibacter symbiosus*, but no further microbial pathogens. The microbiome may have been less divergent since the keratitis occurred without previous traumatic injury; with the result of an uncomplicated clinical course, which was characterised by a rapid improvement after antibiotic administration and supportive therapy.

A rapid and effective treatment of microbial keratitis is of the utmost importance but depends on an accurate identification of all the pathogens involved, which is made possible by a WGS approach. It has already been shown that the antimicrobial treatment was 71% more effective in a group of patients using a 16S rDNA-based NGS approach compared to a control group using conventional culturing methods to identify the pathogens, which led to a significantly better prognosis for these patients [60,61,62]. Based on a study from 2022, the group of Kang concluded that monitoring not only the microbiota but also the antibiotic resistance profiles could lead to significant improvements in the treatment of microbial keratitis, which is possible using WGS instead of 16s rDNA-based NGS [63].

As a “take-home message” of this report, we would like to encourage the introduction of WGS as a diagnostics approach for keratitis. The bioinformatics analysis strategy of a rapid Kraken2-based taxonomy, followed by a confirming Minimap2 back-mapping of the reads to the identified genomes, must be implemented, as it reduces the risk of false-positive classifications. In our view, this diagnostic approach will help to replace the need for multiple species-specific qPCRs in future routine diagnostics enabling an all-encompassing characterisation of polymicrobial communities and potentially their antibiotic resistance genes.

## Figures and Tables

**Figure 1 microorganisms-12-02292-f001:**
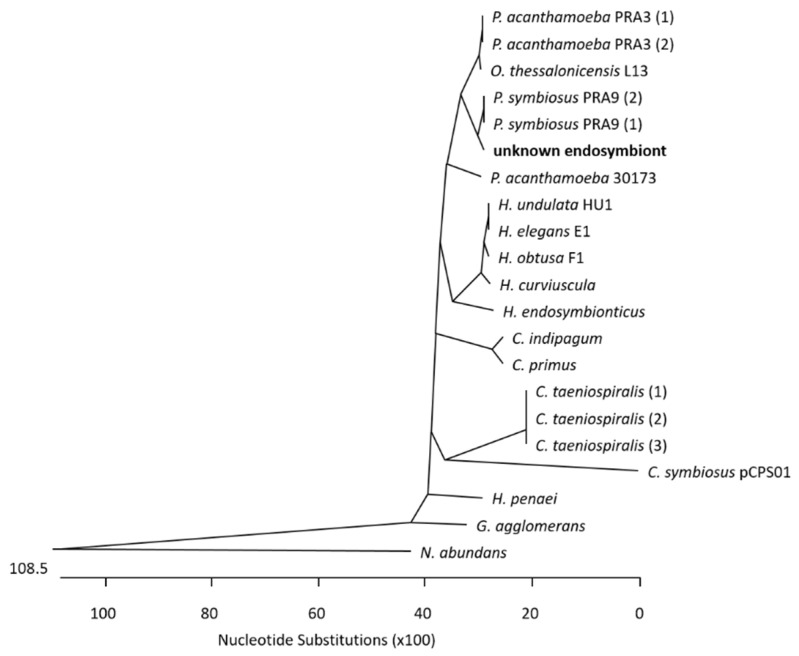
Phylogenetic tree of species of the taxonomic order Holosporales. The sequenced 1.4 kb 16S rDNA region of the unknown endosymbiont was used as a query in Blast analysis, and the most homologous species were used for phylogenetic tree construction by ClustalW method (MegAlign 5.08, DNASTAR, Madison, WI, USA). Accession numbers and abbreviations of the species, shown here, are listed in Appendix A, Sheet Holosporales Species.

**Figure 2 microorganisms-12-02292-f002:**
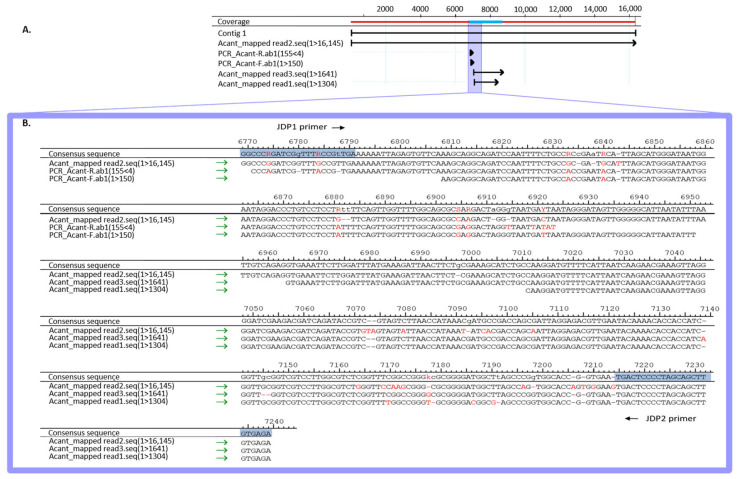
*Acanthamoeba* 18S rDNA sequences. (**A**) Strategy of sequence alignment and (**B**) JDP1-JDP2 18S rDNA region for genotyping. Multiple-sequence alignment of the Sanger sequenced PCR product (PCR_Acant-R.ab1 and PCR_Acant-F.ab1) and Nanopore reads (Acant_mapped read1.seq, Acant_mapped read2.seq, Acant_mapped read3.seq) are shown in JDR1 to JDR2 region. Binding sites for the primers (JDP1 and JDP2) are marked in blue. Single nucleotide differences in the sequences to the consensus are indicated by red letters and deletions by dash.

**Table 1 microorganisms-12-02292-t001:** Microbiome composition of the keratoplasty biopsy.

Organism (NCBI ID)	No. of Reads (%)
*Homo sapiens* (9606)	3,203,564 (99.92%)
*Acanthamoeba castellanii* str, Neff (1257118)	1683 (0.05%)
*Arthrobacter* sp. KBS0702 (2578107)	108 (<0.05%)
*Plasmodium vivax* (5855)	35 (<0.05%)
Candidatus *Paracaedibacter symbiosus* (244582)	32 (<0.05%)

**Table 2 microorganisms-12-02292-t002:** Remapping/Verification of Kraken2 assigned reads with Minimap2.

Reads from Kraken2 Assigned Organism	Mapping Against Reference Genome ^1^	Mapping Against *Homo sapiens* Genome ^1^
*Acanthamoeba castellanii*	933/1683 (55.4%)	831/1683 (49.38%)
*Arthrobacter* sp.	0/108 (0%)	108/108 (100%)
*Plasmodium vivax*	34/35 (97.14%)	35/35 (100%)
Candidatus *Paracaedibacter symbiosus*	29/32 (90.62%)	0/32 (0%)
*Acanthamoeba castellanii*	933/1683 (55.4%)	831/1683 (49.38%)

^1^ Mapped reads/total reads to reference genome (%).

## Data Availability

The original contributions presented in the study are included in the article/Appendix A, further inquiries can be directed to the corresponding author.

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
