# Peer review of "Oxford Nanopore Technology-Based Identification of an *Acanthamoeba castellanii* Endosymbiosis in Microbial Keratitis"

_microorganisms, 2024, doi:10.3390/microorganisms12112292_

Round 1
Reviewer 1 Report
Comments and Suggestions for Authors I consider that the authors present a convenient proposal by suggesting alternative techniques for the detection of amoebas of the genus Acanthamoeba. However, the results they present are not convincing. It would be convenient to present results that undoubtedly allow for a comparison of those obtained with more conventional techniques. It is also convenient that the discussion be enriched by strengthening the central theme of the manuscript, since as proposed, it is not convincing.
Author Response
Comment1: I consider that the authors present a convenient proposal by suggesting alternative techniques for the detection of amoebas of the genus Acanthamoeba. However, the results they present are not convincing. It would be convenient to present results that undoubtedly allow for a comparison of those obtained with more conventional techniques. It is also convenient that the discussion be enriched by strengthening the central theme of the manuscript, since as proposed, it is not convincing.
Response1:
We would like to thank the reviewer for the constructive feedback. We have revised the publication and addressed the reviewers' comments to improve the quality of the manuscript. In the discussion, we have included a section, in which the advantages and disadvantages of the three established methods for Acanthamoeba spp. diagnostics – confocal microscopy, culture of corneal scrapes and PCR - are discussed: “To date three established detection methods for Acanthamoeba in keratitis are discussed in the literature: Culture of corneal scrapings, in vivo confocal microscopy and PCR. Culture is the most conventional method that has been considered the gold standard. However, it was shown to lack on sensitivity and to require a long cultivation period of about 2 weeks, which drastically prolongs the time to diagnosis and other co-pathogens and/or endosymbionts remain undetected or requires additional (cultivation) methods (37-39). Confocal corneal microscopy is significantly faster, but cannot easily differentiate between bacterial and fungal keratitis (16). PCR has been frequently used for the detection of acanthamoebae due to its high sensitivity, specificity and broad range of performance, as demonstrated by the differentiation of Acanthamoeba genotypes (40). When comparing the three methods, different research groups came to different conclusions. Yera et al. compared culture and PCR and found that culture and a single PCR test can lead to a misdiagnosis of Acanthamoeba keratitis, but a combination of different PCR tests could increase the diagnostic sensitivity (41). Two other groups compared in vivo confocal microscopy (IVCM), culture and PCR. They both came to the conclusion that IVCM has the highest sensitivity and specificity in detection (37, 42). However, while Hoffman et al. still considered culture to be the gold standard, Goh et al. instead recommended IVCM or PCR, as it is more readily available compared to IVCM. PCR was also the recommended method compared to culture in a further publication, with which the authors were able to reduce the detection time from approx. 2 weeks to 3 hours (38). The confocal microscopy performed by the department of Ophthalmology at the University Clinic of Duesseldorf clinic had already delivered an initial indication of Acanthamoeba keratitis. On the basis of this clinical suspicion, PCR was carried out at the Institute of Microbiology and Hospital Hygiene, for the detection of Acanthamoeba.” (line 263ff).
In our study, confocal microscopy was already performed by the department of Ophthalmology at the University Clinic of Duesseldorf clinic and provided an initial indication of Acanthamoeba infection. Due to the slow growth times, we opted for PCR detection and compared this established method with the Nanopore-based NGS technology.
To improve the clarity of the manuscript, we have had the publication proofread by a native speaker.
Reviewer 2 Report
Comments and Suggestions for Authors
Report
This study uses a whole genome sequence (WGS) method to identify microbiome constituents in a case of microbial keratitis. Acanthamoeba and bacteria were identified by standard PCR in order to evaluate WGS as a method to more rapidly detect pathogens in sample from the eye surface. This is an important topic and WGS seems to be offer a useful option for Ophthalmologists with access to this facility. The paper is well written and clear, however there are some issues that require further attention.
Acanthamoeba has been identified both by standard PCR and by WGS but no details of the sequences have been made available. The genus Acanthamoeba is comprised of some 23 genotypes defined by the 18S gene T1 to T23, not all of these are equally present in AK cases and T4 is the most frequently identified in AK cases. As the present authors must have the sequence covering the 18S gene by standard PCR and WGS this should be reported and the genotype of the present case determined. Not only will this information be helpful in building a more complete picture of AK but it may be relevant to the further understanding of the relationship between particular Acanthamoeba strains/genotypes and their intracellular pathogen.
Line 2. “Whole Genome Sequencing” should be spelled out in full in the title not just WGS.
Line 17. The word “rather” should be removed.
Like 27. The term “Acanthamoeba Keratitis” should be added to the keywords
Line 94 The target of these primers should be mentioned.
Line 110. The subject of the paper is “Oxford nanopore-based WGS” and so a brief description of the method should be given in the methods.
The supplementary data are uninterpretable without explanation, the spreadsheet columns in the KrakenFullDB_190624 section for example, are not labelled.
Author Response
Comment1: Acanthamoeba has been identified both by standard PCR and by WGS but no details of the sequences have been made available. The genus Acanthamoeba is comprised of some 23 genotypes defined by the 18S gene T1 to T23, not all of these are equally present in AK cases and T4 is the most frequently identified in AK cases. As the present authors must have the sequence covering the 18S gene by standard PCR and WGS this should be reported and the genotype of the present case determined. Not only will this information be helpful in building a more complete picture of AK but it may be relevant to the further understanding of the relationship between particular Acanthamoeba strains/genotypes and their intracellular pathogen
Response1:
We are very grateful for this feedback and ideas, which we have taken into account by including the following additions to the manuscript
- We have uploaded the Nanopore sequencing data to NCBI to make them accessible (BioProject ID: PRJNA1175606).
2. We have identified the genotype of the Acanthamoeba by Blast analysis of a consensus sequence of the 18S rDNA based Acanthamoeba-PCR product and three Nanopore reads, corresponding to the 18S rDNA gene of Acanthamoeba castellanii strain Neff. We have included a respective section and Figure in the publication: “For the genus Acanthamoeba, 23 different genotypes (type T1 to T23) had been identified, defined by differences in the 18S rDNA gene, with T4 as the most frequently identified genotype in AK cases (28). To determine which genotype the detected Acanthamoeba was assigned to, we sequenced the 180 bp 18S rDNA based Acanthamoeba-PCR product of the routine diagnostics, which covered the 5’ end of the JDP1-JDP2 targeted region (that is commonly used for genotyping) (29). Additionally, we mapped the Nanopore sequence reads to the 18S rDNA gene of Acanthamoeba castellanii strain Neff and extracted the three overlapping reads. The consensus sequence of the PCR product and the Nanopore reads of 16,171 bp covered the whole genotyping region JPD1 – JDP2 (see Figure 2). A Blast analysis of this sequence was conducted. Of the possible 23 genotypes only the T4 sequence was found.” (line 207ff).
Comment2: Line 2. “Whole Genome Sequencing” should be spelled out in full in the title not just WGS.
Response2: We have removed the abbreviations from the title. The title has been changed as follows for better readability: “Oxford Nanopore Technology-based identification of an Acanthamoeba castellanii endosymbiosis in microbial keratitis“
Comment3: Line 17. The word “rather” should be removed.
Response3: According to the reviewer’s suggestion, we have removed the word “rather” in the mentioned sentence.
Comment4: Line 27. The term “Acanthamoeba Keratitis” should be added to the keywords
Response4: The missing key word has been added “Acanthamoeba keratitis”. (line 28).
Comment5: Line 94 The target of these primers should be mentioned.
Response5: We have added the target region of the primers used for Acanthamoeba detection to the manuscript: “The inhouse Acanthamoeba qPCR was conducted using primers and probe targeting a 180 bp region of the 18S rRNA gene (listed in Supplementary File 1, Sheet Primers and Probes) as published by Qvarnstrom et al. (25) …“ (line 93ff).
Comment6: Line 110. The subject of the paper is “Oxford nanopore-based WGS” and so a brief description of the method should be given in the methods.
Response6: As suggested by the reviewer we have added a brief description of Oxford Nanopore-based WGS in the methods: “ Oxford Nanopore Technology-based Whole Genome Sequencing (WGS) works by passing single DNA molecules through nanopores, with a constant ionic current. The speed of translocation is controlled by a motor protein that stepwise pushes the nucleic acid molecule through the nanopore. Specific changes in the ion current during translocation correspond to the nucleotide sequence (A, C, G or T) in the sensor region and are decoded using computer algorithms (27).” (line 111ff).
Comment7: The supplementary data are uninterpretable without explanation, the spreadsheet columns in the KrakenFullDB_190624 section for example, are not labelled.
Response7: We apologize for the lack of information. We renamed the section “KrakenFullDB_190624 “ to “KrakenDatabase Version 06-19-24” for reasons of clarity and now have added a column with labels.
Round 2
Reviewer 1 Report
Comments and Suggestions for Authors
I consider that the authors do not highlight the relevance of the study and I do not find any important changes that have enriched their work to improve it.
Author Response
Comment1: I consider that the authors do not highlight the relevance of the study and I do not find any important changes that have enriched their work to improve it.
Response1: In his initial review, the first reviewer stated that the manuscript should be improved by discussing alternative techniques for the detection of acanthamoebae. A corresponding section discussing the currently used standard methods for detection was added. However, as our publication is a work on the microbiome that includes whole genome sequencing techniques, the focus of our manuscript is not on culture or microscopy methods. The relevance of this sequencing method lies in the fact that we detected all relevant organisms with only one analysis approach.